



# Data assimilation for the Model for Prediction Across Scales – Atmosphere with the Joint Effort for Data assimilation Integration (JEDI-MPAS 2.0.0-beta): ensemble of 3D ensemble-variational (En-3DEnVar) assimilations

Jonathan J. Guerrette[1], Zhiquan Liu[1], Chris Snyder[1], Byoung-Joo Jung[1], Craig S. Schwartz[1],
Junmei Ban[1], Steven Vahl[1,3], Yali Wu[1,4], Ivette Hernández Baños[1,5], Yonggang G. Yu[1,6], Soyoung Ha[1],
Yannick Trémolet[2], Thomas Auligné[2], Clementine Gas[2], Benjamin Ménétrier[2], Anna Shlyaeva[2],
Mark Miesch[2,7], Stephen Herbener[2], Emily Liu[2,8], Daniel Holdaway[2,9], and Benjamin T. Johnson[2]

[1]National Center for Atmospheric Research, Boulder, Colorado 80301, USA
[2]Joint Center for Satellite Data Assimilation, Boulder, Colorado 80301, USA
[3]Now at the Joint Center for Satellite Data Assimilation, Boulder, USA
[4]Now at the Shenzhen Institute of Meteorological Innovation, Shenzhen, China
[5]Now at the Scripps Institution of Oceanography at UC San Diego, La Jolla, USA
[6]Now at the Science Applications International Corporation, Reston, USA
[7]Now at CIRES at University of Colorado and NOAA Space Weather Prediction Center, Boulder, USA
[8]Now at the NOAA National Centers for Environmental Prediction, College Park, USA
[9]Now at the NASA Goddard Space Flight Center, Greenbelt, USA

**Correspondence:** Jonathan Guerrette (guerrett@ucar.edu)

**Abstract.** An ensemble of three-dimensional ensemble-variational (En-3DEnVar) data assimilations is demonstrated with the Joint Effort for Data assimilation Integration (JEDI) with the Model for Prediction Across Scales – Atmosphere (MPAS-A) (i.e., JEDI-MPAS). Basic software building blocks are reused from previously presented deterministic 3DEnVar functionality, and combined with a formal experimental workflow manager in MPAS-Workflow. En-3DEnVar is used to produce an 80-
member ensemble of analyses, which are cycled with ensemble forecasts in a 1-month experiment. The ensemble forecasts approximate a purely flow-dependent background error covariance (BEC) at each analysis time. The En-3DEnVar BECs and prior ensemble mean forecast errors are compared to those produced by a similar experiment that uses the Data Assimilation Research Testbed (DART) Ensemble Adjustment Kalman Filter (EAKF). The experiment using En-3DEnVar produces similar ensemble spread to and slightly smaller errors than the EAKF. The ensemble forecasts initialized from En-3DEnVar and EAKF
analyses are used as BECs in deterministic cycling 3DEnVar experiments, which are compared to a control experiment that uses 20-member MPAS-A forecasts initialized from Global Ensemble Forecast System (GEFS) initial conditions. The experimental ensembles achieve mostly equivalent or better performance than the off-the-shelf ensemble system in this deterministic cycling setting; although, there are many obvious differences in configuration between GEFS and the two MPAS ensemble systems. An additional experiment that uses hybrid 3DEnVar, which combines the En-3DEnVar ensemble BEC with a climatological
BEC, increases tropospheric forecast quality compared to the corresponding pure 3DEnVar experiment. The JEDI-MPAS En-3DEnVar is technically working and useful for future research studies. Tuning of observation errors and spread is needed to



improve performance and several algorithmic advancements are needed to improve computational efficiency for larger-scale applications.

# 1 Introduction

Liu et al. (2022) introduced a new data assimilation (DA) system for the Model for Prediction Across Scales - Atmosphere (MPAS-A; Skamarock et al., 2012) that is built on the Joint Effort for Data assimilation Integration (JEDI; Trémolet and Auligné, 2020) software framework, called JEDI-MPAS. Liu et al. (2022) demonstrated JEDI-MPAS for global three-dimensional ensemble variational (3DEnVar) DA to produce a deterministic (single state) analysis of the atmosphere. When approximating flow-dependent background error covariances (BEC) in 3DEnVar, they used 20-member ensembles of MPAS-A forecasts that

were initialized from initial conditions used by the National Centers for Environmental Prediction (NCEP) Global Ensemble Forecast System (GEFS) (Zhou et al., 2017).

It would be preferable for a myriad of research applications that the ensemble used in the DA be generated by JEDI-MPAS itself, so that it is consistent with the details–such as the characteristics of the observing network–of a given application. Here we evaluate using an ensemble of data assimilations (EDA), implemented generically in JEDI and available for any

forecast model with JEDI interfaces, to generate ensemble initial conditions for JEDI-MPAS. Our first EDA implementation is for an ensemble of 3DEnVars (En-3DEnVar), because it requires very few modifications to the EnVar algorithm previously described by Liu et al. (2022). A major attraction of EDA based on variational algorithms is that most technical or algorithmic improvements targeted for deterministic DA will directly translate to the ensemble system.

The technique that we term EDA involves conducting an ensemble of independent analysis and forecast steps in each

cycle. Each member ingests perturbed versions of the available observations during its assimilation step. This technique was proposed and tested by Houtekamer and Derome (1995), who called the approach "OSSE-MC." Subsequent studies employed EDA (under other names) to generate analysis ensembles for ensemble forecasting systems (Houtekamer et al., 1996; Hamill et al., 2000), or for estimating forecast-error statistics for DA covariance modeling (Fisher, 2003; Zagar et al., 2005; Berre et al., 2006), or in "stochastic" ensemble Kalman filters (EnKF) (Houtekamer and Mitchell, 1998; Burgers et al., 1998, and

many subsequent papers).

Variational-based EDA was first implemented operationally as an ensemble of four-dimensional variational (En-4DVar) minimizations at Météo-France (Berre et al., 2007; Desroziers et al., 2008; Berre and Desroziers, 2010), and then at the European Centre for Medium-Range Weather Forecasts (Isaksen et al., 2010). The UK Met Office later replaced its Local Ensemble Transform Kalman Filter (LETKF) with an ensemble of hybrid 4DEnVar's (En-4DEnVar) in their ensemble prediction system

through extensive efforts (Bowler et al., 2017a, b; Lorenc et al., 2017) and following multiple motivations: code maintenance is reduced via shared software with their deterministic 4DVar; there is an improved capability to use more advanced model-space localization techniques (i.e., Lorenc, 2017); the En-4DEnVar produced faster and more realistic ensemble spread growth in forecasts than LETKF (Bowler et al., 2017b); En-4DEnVar perturbations used as flow-dependent BECs improved forecasts in





their deterministic hybrid 4DVar system compared to LETKF perturbations (Bowler et al., 2017a); and En-4DEnVar is cheaper

than an alternative En-4DVar.

There are numerous techniques besides variational-based EDA for generating ensembles of initial conditions, including 4D ensemble Kalman Smoothers (Evensen, 2003). In particular, a relatively mature EnKF for MPAS exists (Ha et al., 2017), which is based on the Data Assimilation Research Testbed (DART; Anderson et al., 2009); we will call this system MPAS-DART. Although not a focus of this paper, MPAS-DART EnKF is a useful benchmark for initial evaluations of the JEDI-MPAS

En-3DEnVar and we present companion results from MPAS-DART for many aspects of JEDI-MPAS performance.

While variational-based EDA poses a large potential benefit of increased skill via re-use of 4D algorithms, it also incurs more computational overhead per member than many EnKF algorithms. There are numerous EDA algorithm advances in previous works to alleviate some of the added cost compared to EnKF, such as separating the update of the ensemble mean and perturbations and using simpler, cheaper configurations for the perturbation update (Buehner et al., 2017; Lorenc et al.,

2017), block minimization methods (Mercier et al., 2018), and advanced in-memory storage and communication strategies for the numerous ensemble perturbations (Arbogast et al., 2017). We do not consider such enhancements in this paper, though they may certainly be helpful in the future.

The outline of the paper is as follows. In Sect. 2 we briefly describe the JEDI-MPAS En-3DEnVar implementation and the MPAS-DART EnKF, both of which are used in our experiments. In Sect. 4, we compare the 6 h ensemble forecast statistics

produced with the JEDI-MPAS En-3DEnVar and MPAS-DART EnKF across one month of cycling on the MPAS-A quasi-uniform 60 km mesh. Although we do not present a general comparison of EnKF and EDA, there are very few comparisons in literature (e.g., Hamrud et al., 2015; Bonavita et al., 2015), and this work gives another data point. In Sect. 5, we use the 6 h forecasts initialized from MPAS-DART EnKF and JEDI-MPAS EDA analyses as ensemble BECs in several dual-mesh "30km-60km" (i.e., 30 km outer loop and 60 km inner loops) 3DEnVar deterministic cycling experiments to show the utility of the

EDA in producing flow-dependent BECs. Finally we finish with conclusions and a future outlook for JEDI-MPAS ensemble DA in Sect. 6.

## 2 Ensemble data assimilation

### 2.1 JEDI-MPAS EDA

Liu et al. (2022) described the 3DEnVar algorithm implemented in JEDI, and thus JEDI-MPAS, following Lorenc (2003) and

Buehner (2005). We utilize the same algorithm and software implementation in the En-3DEnVar and in subsequent deterministic 3DEnVar experiments. In the EDA there are $N_e$ independent cost functions, where the $i$th EDA cost function is evaluated at the $i$th background state $\boldsymbol{x}_{b,i}$,

$$J_i(\boldsymbol{x}) = \frac{1}{2}(\boldsymbol{x} - \boldsymbol{x}_{b,i})^{\mathbf{T}}\mathbf{B}_i^{-1}(\boldsymbol{x} - \boldsymbol{x}_{b,i}) + \frac{1}{2}[h(\boldsymbol{x}) - \boldsymbol{y} - \boldsymbol{\epsilon}_i]^{\mathbf{T}}\mathbf{R}^{-1}[h(\boldsymbol{x}) - \boldsymbol{y} - \boldsymbol{\epsilon}_i]. \qquad (1)$$

The vectors $\boldsymbol{\epsilon}_i$ are realizations of the random observation error $\boldsymbol{\epsilon} \sim N(0, \mathbf{R})$, where $\mathbf{R}$ is the observation error covariance

matrix and the ensemble mean of the realizations is removed so that $\sum \boldsymbol{\epsilon}_i = 0$. The observation operator, $h$, simulates model-





equivalent observations given the state, $x$. In each outer iteration of a truncated Gauss-Newton minimization (Lawless et al., 2005), an inner loop minimization uses a linear approximation of Eq. 1 to determine the analysis increment. The increment is added to the current guess of $x$, beginning at $x_{b,i}$ in the first outer iteration.

As described by Liu et al. (2022), the $i$th BEC, $\mathbf{B}_i$ in Eq. 1, is a weighted sum of the climatological background error
covariance $\mathbf{B}_c$ and the member-specific sample ensemble covariance $\mathbf{B}_{e,i}$, i.e.,

$$\mathbf{B}_i = \beta_c \mathbf{B}_c + \beta_e \mathbf{L} \circ \mathbf{B}_{e,i}, \tag{2}$$

where $\beta_c$ and $\beta_e$ are scalar weights with $\beta_c + \beta_e = 1$. $\beta_c$, $\beta_e$, and $\mathbf{B}_c$ are identical for all EDA members. $\mathbf{L} \circ \mathbf{B}_{e,i}$ denotes the Schur product (element-by-element) of the localization matrix $\mathbf{L}$ and $\mathbf{B}_{e,i}$. The member-specific $\mathbf{B}_{e,i}$ allows for self-exclusion, described in Sect. 2.1.1. Note that $\mathbf{L}$ is a correlation matrix with diagonal elements being 1 and off-diagonal elements
smaller than 1 that reduce to zero for a certain distance between two model grid points. Therefore, the localization matrix reduces spurious correlations in $\mathbf{B}_{e,i}$ caused by sampling errors associated with a limited ensemble size. The only difference between the description above for EDA cycling and deterministic cycling is that the latter only has one background state, the observations are not perturbed, and the BEC only has one realization based on all $N_e$ members.

As described by Liu et al. (2022), the analysis variables are temperature ($T$), horizontal wind components ($U$, $V$), surface
pressure ($P_s$), and specific humidity ($Q_v$). One difference from Liu et al. (2022) is that the hydrostatic balance constraint in the analysis increment stage is no longer applied directly to the full analysis variables. Instead, an incremental form of the hypsometric equation is used to approximate the dry air density ($\rho_d$) and 3-d pressure ($P$) increments from the increments in $T$, $P_s$, and $Q_v$. The hypsometric equation is linearized around a hydrostatic state constructed using the previous outer iteration analysis, $x_i$, of $T$, $P_s$, and $Q_v$. After the DA minimization is complete, the analysis state is transformed to the MPAS-A
prognostic variables during the model initialization before the model's time integration.

The ensemble BEC, $\mathbf{B}_e$, is represented by prior perturbations (before assimilation) in those same analysis variables with respect to the prior ensemble mean. As described in Jung and et al. (2023), $\mathbf{B}_c$ is constructed by applying linear transformations that yield the analysis variables from stream function, velocity potential, and the "unbalanced" contributions to temperature and surface pressure, together with the assumption that background errors in those underlying variables are mutually independent
and have known, isotropic covariances (Derber and Bouttier, 1999). $\mathbf{B}_c$ is implemented via generic JEDI interfaces to the Background error on Unstructured Mesh Package (BUMP; Ménétrier, 2020).

### 2.1.1 Self-exclusion

As first shown by Houtekamer and Mitchell (1998), updating an ensemble of forecasts using an assimilation scheme based on the sample covariances of that same ensemble, as in En3DEnVar for example, leads to an analysis ensemble with too little
spread when compared to the errors of the analysis mean. To counteract this systematic bias in the update, they proposed splitting the ensemble into subsets and updating members in a given subset using the sample covariance from members in the other subsets.





In the limit that the subsets contain a single member, each member $i$ in the EDA will use in the cost function (1) a different, flow-dependent BEC $\mathbf{B}_{e,i}$, obtained by omitting $\boldsymbol{\delta x}_i = \boldsymbol{x}_i - \bar{\boldsymbol{x}}_b$, the $i$th ensemble perturbation, from the computation of $\mathbf{B}_{e,i}$,

where $\bar{\boldsymbol{x}}_b$ is the ensemble mean. Sacher and Bartello (2008) and Mitchell and Houtekamer (2009) showed with small toy problems that this approach causes the posterior ensemble spread to overestimate the the root-mean-square error (RMSE) of the posterior ensemble mean.

Bowler et al. (2017b) called the removal of $\boldsymbol{\delta x}_i$ from $\mathbf{B}_{e,i}$ "self-exclusion", applying it to an En-4DEnVar, while Buehner (2020) called it "cross-validation", applying it to an LETKF. Bowler et al. (2017b) and Buehner (2020) both found that self-

exclusion reduced the spread reduction that occurred during the DA procedure going from background ensemble to the analysis ensemble. Even so, Bowler et al. (2017b) found that applying self-exclusion necessitated the use of more relaxation than when not using self-exclusion (see Sect. 2.1.2 for a definition of relaxation). Self-exclusion is applied in the JEDI-MPAS EDA experiment described in Sect. 4.

### 2.1.2 Ensemble spread tuning

The EDA update described above will systematically underestimate the analysis uncertainty to some degree, despite employing multiple techniques to reduce the detrimental effects of sampling error: covariance localization in the ensemble BEC, hybridization with a static BEC, and self-exclusion. More important, the ensemble forecasts in JEDI-MPAS do not at present account for model error, so even if the analysis ensemble is perfectly representative of the statistics of analysis error the ensemble forecast will be underdispersive. Both effects will also accumulate over successive forecast-analysis cycles. For these

reasons, it is essential that JEDI-MPAS include some method for tuning the overall ensemble spread.

There are many approaches for tuning ensemble spread to ensure stable cycling of the ensemble-dependent DA and forecast system. In the Relaxation to Prior Perturbations (RTPP; Zhang et al., 2004) method, the analysis perturbation for member $i$, $\boldsymbol{\delta x}_{a,i}$, is replaced by a weighted sum of $\boldsymbol{\delta x}_{a,i}$ and $\boldsymbol{\delta x}_{b,i}$ with a scalar weight $\alpha_{\mathrm{RTPP}}$, i.e.,

$$\boldsymbol{\delta x}_{a,i} \leftarrow (1 - \alpha_{\mathrm{RTPP}})\boldsymbol{\delta x}_{a,i} + \alpha_{\mathrm{RTPP}}\boldsymbol{\delta x}_{b,i}. \tag{3}$$

Thus, the relaxed ensemble perturbations take on some of the observationally constrained analysis perturbations, $\boldsymbol{\delta x}_{a,i}$, and the forecast model-driven background perturbations, $\boldsymbol{\delta x}_{b,i}$. In JEDI-MPAS, RTPP is carried out via a stand-alone executable, one which inherits from a generic implementation in the JEDI Object-Oriented Prediction System (OOPS) for the *RTPP* application.

Whitaker and Hamill (2012) proposed an alternative to RTPP called Relaxation to Prior Spread (RTPS), which relaxes the

spread of the analysis ensemble toward the background ensemble spread, instead of relaxing the perturbations. The underpinning of RTPS is the spread change ratio, $\boldsymbol{s}$, whose $j$th element, associated with a given grid cell and analysis variable, is calculated as

$$s_j = \frac{\sigma_{b,j} - \sigma_{a,j}}{\sigma_{a,j}}, \tag{4}$$





where $\sigma_{b,j}$ and $\sigma_{a,j}$ are the prior and posterior ensemble sample standard deviations (spreads). RTPS operates independently
for each $j$ using a Schur product,

$$\boldsymbol{\delta x}_{a,i} \leftarrow \boldsymbol{\delta x}_{a,i} \circ (\alpha_{\mathrm{RTPS}} \boldsymbol{s} + \mathbf{1}), \tag{5}$$

It is common practice to combine RTPP with RTPS (e.g., Bowler et al., 2017b), or with multiplicative inflation, or to adapt $\alpha$ at
each DA cycle (e.g., Kotsuki et al., 2017). Here we only use RTPP with a fixed global $\alpha$ throughout all DA cycles as a means
of providing an initial JEDI-MPAS EDA functionality to the community. There remain many opportunities for improving
ensemble spread in future work.

### 2.1.3   Implementation

In the EDA experiment presented here, each member is treated with a fully independent execution of the JEDI-MPAS *Varia-tional* application (https://jointcenterforsatellitedataassimilation-jedi-docs.readthedocs-hosted.com/en/latest/inside/jedi-components/oops/applications/variational.html). OOPS contains implementations for generic model-independent applications in JEDI. The
generic OOPS *Variational* application can be used to conduct 3DVar, 3DEnVar, 4DEnVar, and 4DVar (for models with lin-earized tangent and adjoint descriptions), as well as applicable hybrid variants thereof. Only 3DVar, 3DEnVar, 4DEnVar, and
hybrid variants are enabled in the JEDI-MPAS model-specific *Variational* executable at this time. Self-exclusion is achieved
in En-3DEnVar simply by removing each members' own background state from a list of ensemble members in the *Variational*
application configuration.

The ensemble forecasts and EDA are conducted via our open-source MPAS-Workflow (https://github.com/NCAR/MPAS-Workflow),
which uses the Cylc general purpose workflow manager (Oliver et al., 2019) v7.8.3 to orchestrate tasks written in a combination
of c-shell and python scripts. MPAS-Workflow automatically constructs *Variational* application configuration files in YAML
format for each cycle. At the time of writing, MPAS-Workflow only operates on the Cheyenne HPC managed by NCAR's
Computational and Information Systems Laboratory (CISL). MPAS-Workflow handles a small set of use-cases specific to the
NCAR Microscale and Mesoscale Meteorology (MMM) Laboratory. Although it is not yet designed for general purpose use,
this open-source repository might serve to instruct others on how to run JEDI-MPAS and MPAS-A together.

The source code used for our experiments are provided in the JEDI-MPAS 2.0.0-beta release version, as described in the
code availability section.

### 2.2   EAKF in MPAS-DART

To assess the credibility of our newly developed JEDI-MPAS EDA system, an ensemble adjustment Kalman filter (EAKF;
Anderson, 2001, 2003; Anderson and Collins, 2007) implemented within the "Manhattan" version of DART (Anderson et al.,
2009) is also used to produce analyses. DART is a mature software platform for ensemble-based DA and has been inter-faced with MPAS-A (Ha et al., 2017). DART can perform both stochastic and deterministic EnKF algorithms, where only the
former perturbs observations. When background and observation error distributions are near Gaussian, the use of perturbed
observations is known to degrade the quality of the ensemble-mean analysis relative to that produced by deterministic filters



(Anderson, 2001; Whitaker and Hamill, 2002) and a theoretically similar deterministic variational-based EDA (Bowler et al., 2012). On the other hand, as the forecast model and observation operators become increasingly nonlinear, there is evidence that an EDA with perturbed observations avoids some pathological behaviors that appear in deterministic EnKFs (Lawson and Hansen, 2004; Anderson, 2010; Lei et al., 2010; Anderson, 2020).

As a deterministic square-root variant of the EnKF, the EAKF has many differences from the EDA algorithm described in Sect. 2.1. The primary difference is that EAKF does not perturb observations; all members assimilate an identical realization of a given observation (i.e., the measured observation). Another technical difference is that the EAKF assimilates observations one-at-a-time, whereas variational minimizations assimilate all observations simultaneously. Moreover, while 3DEnVar applies covariance localization in model space, the EAKF applies localization to observation-observation and observation-state

covariances, which may be suboptimal for assimilation of radiance observations (Campbell et al., 2010; Lei et al., 2018). Additionally, the EAKF does not employ self-exclusion (Sect. 2.1.1) to compute unique BECs for each ensemble member, and there are also differences regarding posterior relaxation between EDA and the EAKF (described in Sect. 4.1). Finally, MPAS's interface with DART does not use a hydrostatic pressure constraint on analysis increments, unlike JEDI-MPAS's variational algorithm (Sect. 2.1).

There are clearly many differences between the En-3DEnVar implemented in JEDI-MPAS and the EAKF in MPAS-DART. Moreover, neither system has been thoroughly tuned and DART has many capabilities that we do not exercise, including sophisticated inflation and localization options. We therefore do not attempt to attribute differences between their performances to specific settings or parameters. Instead, we view MPAS-DART as a convenient baseline against which to compare the robustness and validity of our newly developed EDA implementation within JEDI-MPAS.

## 195    3    Model and observation configurations

### 3.1    MPAS-A model

MPAS-A is a non-hydrostatic model discretized on an unstructured centroidal Voronoi mesh in the horizontal with C-grid staggering of the state variables, and works for both global and regional applications (Skamarock et al., 2012, 2018). Herein we present results using two different MPAS-A quasi-uniform meshes, 60 km (163,842 horizontal columns) and 30 km (655,362

columns). All time integrations for the 60 km mesh use a 360 s time step, while those for the 30 km mesh use a 180 s time step. Additional sensitivity experiments are described that utilized the quasi-uniform 120 km mesh (40,962 columns) with a 720 s time step. All meshes utilize 55 vertical levels with a 30 km model top and the "mesoscale reference" physics suite, as described by Liu et al. (2022).

In almost all respects, the same modified version of MPAS-A version 7.1 that was used by Liu et al. (2022) is used here.

Some minor code modifications are included in the JEDI-MPAS 2.0.0-beta release version, as described in the code availability section.





## 3.2  Observations

All experiments assimilate the same set of observations. We convert netCDF-formatted observation diagnostic files from the Gridpoint Statistical Interpolation (GSI; Shao et al., 2016) system (i.e., "GSI-ncdiag") to a format that can be read by the JEDI Interface for Observation Data Access (IODA; Honeyager et al., 2020). For in situ observations, we assimilate sondes (temperature, virtual temperature, zonal and meridional wind components, specific humidity), aircraft (temperature, zonal and meridional wind components, specific humidity), and surface pressure. For non-radiance remote observations, we assimilate satellite atmospheric motion vectors (AMV, zonal and meridional wind components) and Global Navigation Satellite System and Global Positioning System Radio Occultation (collectively referred to as GNSSRO herein) refractivity.

We assimilate clear-sky microwave radiances as brightness temperature from 6 Advanced Microwave Sounding Unit-A (AMSU-A) sensors aboard NOAA-15, NOAA-18, NOAA-19, AQUA, METOP-A, and METOP-B. We only assimilate channels 5 to 9, because higher peaking channels are sensitive to stratospheric regions that are above the 30 km model top, and thus cannot be simulated correctly. Additionally, some AMSU-A channels are removed following sensitivity experiments that showed larger RMSEs or lower quality in 6 h forecasts: NOAA-19 channel 8; AQUA channels 5, 6, and 7; METOP-A channels 7 and 8; and METOP-B channels 5, 6, and 7.

The GSI-ncdiag files include brightness temperature bias correction, which is calculated using variational bias correction (VarBC) in GSI to correct for fluctuations in instrument bias. We add those pre-computed bias corrections directly to the observed brightness temperature before reading the IODA-formatted observations into JEDI-MPAS. Similarly, observation error standard deviations (square-root of diagonal of $\mathbf{R}$) come directly from the GSI-ncdiag files for most observation types. The only exceptions are for GNSSRO and satellite AMVs. The GNSSRO refractivity errors are calculated online within the JEDI Unified Forward Operator (UFO; Honeyager et al., 2020) using a height-dependent parameterization ported from GSI.

In early sensitivity experiments, we determined that the observation errors for AMVs provided in GSI-ncdiag files were much larger than the RMSE of JEDI-MPAS background innovations, $d_j = h_j(\boldsymbol{x}_b) - y_j$, and that the observations were much denser than either of the 30 km or 60 km meshes. We opt to reduce the correlations between dense observations by thinning them horizontally. Both AMVs and radiances are thinned on a 145 km global Gaussian mesh. Also, we opt to decrease the prescribed AMV observation errors according to the pressure-dependent values shown in Table 1, with linear interpolation between the pressures shown, and following the same parameterization as an early JCSDA near-real-time prototype (personal communications with Greg Thompson). As will be discussed in the context of the results, there are many opportunities yet to optimize the observation errors for the JEDI-MPAS cycling system, but that is not the major focus of this work.

We use a quality control (QC) check for all observations that allows for maximum "PreQC" quality flags (as provided in the GSI-ncdiag files) of 0 and 3 for radiance and non-radiance instruments, respectively. PreQC includes various checks for raw data quality, as well as background innovation checks from GSI based on its own background state. We additionally filter observation locations and variables that exceed a $3\sigma_o$ background check (i.e., the observation-error normalized absolute innovation must satisfy $\frac{|d|}{\sigma_o} \leq 3$ to be assimilated). Surface pressure locations are removed when the model elevation and





observing station elevation differ by more than 200 m. Also, the surface pressure forward operator includes a height correction
following the appendix of Ingleby (2013).

## 4 Ensemble cycling

### 4.1 Setup

We conduct two ensemble cycling experiments with nearly identical settings. One experiment uses the JEDI-MPAS EDA
(*EDA*) and the other experiment uses the MPAS-DART EAKF (*DART*). Both experiments use 80 initial ensemble backgrounds
generated by integrating 4 sets of 20 MPAS-A forecasts. The 4 forecast sets are initialized from 20-member GEFS initial
conditions (i.e., 0 h forecasts) valid at 00:00 UTC, 06:00 UTC, 12:00 UTC, and 18:00 UTC 14 April 2018. Thus the first cycle
background ensemble at 00:00 UTC 15 April 2018 is comprised of 24 h, 18 h, 12 h, and 6 h forecasts on the 60 km mesh.

Both ensemble experiments alternate between data assimilation and an ensemble of 6 h forecasts at each cycle until 18:00
UTC 14 May 2018, ending with a final forecast valid at 00:00 UTC 15 May 2018. The results that follow in Sect. 4.2 reflect
statistics calculated from 00:00 UTC 17 April 2018 to 18:00 UTC 14 May 2018 (inclusive) to allow 2 days for spin-up of
characteristic errors. The unique aspects of the *EDA* and *DART* experiments follow.

Each *EDA* variational minimization uses a single outer loop iteration and 60 inner loop iterations. The ensemble BEC local-
ization uses fixed length scales of 1200 km and 6 km in horizontal and vertical dimensions, respectively. Prior to conducting
the 60 km experiments presented here, we carried out 1-month EDA sensitivity experiments on the 120 km mesh to deter-
mine the impacts of self-exclusion and RTPP. We found that using self-exclusion increased background ensemble spread for a
fixed $\alpha_{\mathrm{RTPP}}$ and improved forecast verification scores compared to not using self-exclusion. Therefore, we use self-exclusion
without further sensitivity study in the 60 km *EDA* experiment.

Among selected $\alpha_{\mathrm{RTPP}}$, varying from 0.5 to 0.95 in 0.05 steps, we found that $\alpha_{\mathrm{RTPP}} = 0.80$ yielded the best 1-to-10-
day forecasts in 120 km sensitivity tests with 20 ensemble members. Those forecasts were initialized by the mean of the
20-member analysis ensembles at 00:00 UTC and 12:00 UTC from 15 April 2018 to 4 May 2018. We also generally found
improvement when increasing ensemble size from 20 members to 80 members while using $\alpha_{\mathrm{RTPP}} = 0.80$, with some saturation
of forecast quality seen with only 40 members. As Bowler et al. (2017b) described, $\alpha_{\mathrm{RTPP}}$ can be decreased when increasing
ensemble size, because sampling error is reduced. Similarly, inflationary measures can be reduced when decreasing mesh
spacing, because sub-grid-scale model error is reduced, mitigating under-dispersiveness in ensemble forecasts. Therefore we
executed two different 20-member 10-day EDA experiments on the 60 km mesh with $\alpha_{\mathrm{RTPP}} = 0.4$ and $\alpha_{\mathrm{RTPP}} = 0.7$. Their
mean background RMSE was much less sensitive to the relaxation coefficient than were the 20-member experiments on the
120 km mesh. Therefore, we chose $\alpha_{\mathrm{RTPP}} = 0.7$ for the 80-member 60 km *EDA* experiment, even if it may not be an optimal
setting.

The *DART* experiment is identical to the *EDA* experiment in terms of initialization, cycling period (6 hours), and cycling
duration. In addition, the EAKF uses the same 1200 km horizontal and 6 km vertical localization lengths scales as *EDA*,





although localization in *DART* is in observation-space rather than model-space (see Sect. 2.2). *DART* uses RTPS with $\alpha_{\mathrm{RTPS}} =$ 1.0 to maintain ensemble spread throughout cycling.

For observations, *DART* uses the JEDI-MPAS model-specific implementation of the OOPS *HofX3D* application (https://

jointcenterforsatellitedataassimilation-jedi-docs.readthedocs-hosted.com/en/latest/inside/jedi-components/oops/applications/hofx. html) to apply forward operators for each ensemble member, assign observation errors, and perform quality control and observation thinning. The output of *HofX3D* is then ingested into MPAS-DART, bypassing DART's own forward operators and observation processing capabilities.

Given the coupling of *HofX3D* to MPAS-DART, both *EDA* and *DART* utilize identical observations possessing and identical

observation errors (Sect. 3.2). However, there is one subtle difference regarding observation QC filtering. The background check for all members in *DART* is based on the prior ensemble mean, whereas the background check in *EDA* is applied independently to each ensemble member before the observations are perturbed. That difference, coupled with differences in algorithm and inflationary measures between *EDA* and *DART* (Sect. 2), mean that throughout the month of cycling the two experiments assimilate slightly different observations, because different observations could fail the background check between experiments.

Nonetheless, any differences in assimilated observations reflect differences in the assimilation algorithms themselves and the quality of the assimilation systems.

Throughout this section and Sect. 5, comparisons are made to initial conditions (referred to herein as "analyses") used in the NCEP Global Forecasting System (GFS). The GFS analyses are transformed to the same mesh used to produce forecasts (e.g., 60-km or 30-km) via MPAS-A's initialization procedures. GFS is a well-tuned operational forecast system initialized from a

deterministic analysis produced by NCEP's Global Data Assimilation System (GDAS) hybrid 4DEnVar (Kleist and Ide, 2015). Thus, we expect GFS analyses to be more accurate than the analyses produced in our own experiments.

## 4.2 Results

If GFS analyses are considered as truth, and the MPAS ensemble background states are unbiased relative to that truth, then the optimal background ensemble spread ($\sigma_{x_b}$) is equal to the RMSE of differences between the prior ensemble mean and

GFS analyses (i.e., rms($\delta x_{GFSa}$)), averaged over a sufficient number of valid times. The MPAS ensemble background is not unbiased with respect to GFS analyses, especially near the model top for temperature ($T$), zonal wind ($U$), and pressure ($P$) (not shown). Although the operational GFS analyses are undoubtedly more accurate than the JEDI-MPAS ensemble mean backgrounds, they are still not equal to the truth. RMSEs of longer duration ensemble forecast mean states with respect to independent analyses are useful for diagnosing spread growth characteristics (e.g., Bowler et al., 2017b), but such measures at

a 6-hour forecast length should be considered qualitative.

With those caveats in mind, Fig. 1 shows rms($\sigma_{x_b}$) and rms($\delta x_{GFSa}$) for *DART* and *EDA*, aggregated over all horizontal columns and varying with model level. There is a strong indication that the background ensemble spreads are too small or the RMSE is too large, especially near the top of the model and near the surface for $T$. Since we have not employed any measures to account for model uncertainty (see Sect. 2.1.2), we expect that the ensemble forecasts will be under-dispersive. Overall,

*DART* and *EDA* produce similar ensemble spreads and ensemble mean RMSEs at most model levels.





Figure 2 dissects the same quantities for model simulated $P_s$, varying with latitude. There are obvious zonal differences between the two ensemble cycling experiments. *DART* produces slightly larger $P_s$ spread than *EDA* in the tropics and smaller spread elsewhere. Those spread variations do not correlate directly with RMSE, since *EDA* produces a smaller RMSE at all latitudes. In separate diagnostics that further narrow down the $P_s$ spread and RMSE on a world map (not shown), we found that *DART*'s larger local spread produces lower RMSE in the western tropical Pacific Ocean and *EDA*'s smaller local spread produces lower RMSE in the tropical Atlantic Ocean and southeastern Pacific Ocean. Those responses could be associated with the relative availabilities of local $P_s$ observations, with there being slightly more available in the western Pacific.

The *DART* experiment has diminishing $U$ and $V$ ensemble spread in the tropical free troposphere as the cycling progresses (not shown). That spread loss could be caused by previously documented characteristics of RTPS, which Bowler et al. (2017b) found to produce inflation at much smaller scales than RTPP. Thus, that behavior is not indicative of relative performances of EAKF and EDA algorithms. The full investigation is reserved for future work, since the *DART* experiment is not the target of our current effort. However, the differences in $U$ and $V$ ensemble spread between *DART* and *EDA* have a non-negligible impact on the deterministic results presented in Sect. 5.2.

In fact, the ensemble spread decrease is likely the cause for differences in innovation RMSE for satellite AMVs between *EDA* and *DART* above 650 hPa (Fig. 3). Also shown are the total spread (Andersson et al., 2003; Desroziers et al., 2005) for both experiments. From Fig. 3, one might conclude that the observation error and/or the ensemble spread are too small to account for the ensemble mean RMSE. After running all of the experiments in this study, we found that GSI assigns unique observation errors for each satellite AMV data source (e.g., Geostationary Operational Environmental Satellite (GOES), European Organisation for the Exploitation of Meteorological Satellites (EUMETSAT), Japan Meteorological Agency (JMA)) and instrument band (e.g., infrared window, infrared water vapor, visible). Therefore the single vertical error distribution assigned in Table 1 for all AMVs needs to be revisited.

All things considered, the *EDA* and *DART* experiments produce remarkably similar behaviors. Those similarities are attributable to their commonalities in settings used for observation processing and lack of tuning. There are yet many opportunities to reduce the prior ensemble mean RMSE for both experiments, including observation error tuning, fixing known issues in GNSSRO assimilation (see Sect. 6), and assimilating more observation types. There are also many opportunities to increase the ensemble spread such that the consistency between spread and RMSE is improved, including accounting for model uncertainty, tuning the relaxation mechanism(s), and applying additive and prior inflation.

# 5 Deterministic cycling

## 5.1 Setup

Both the *EDA* and *DART* ensemble cycling experiments successfully cycled for an entire month. They exhibit nearly stable spread characteristics, even if the spread tends to be too narrow relative to ensemble mean RMSEs. Therefore, their respective background ensemble forecasts might be effective in deterministic 3DEnVar cycling experiments. We conduct four deterministic dual-mesh 30km-60km 3DEnVar cycling experiments, where the 30km mesh is used in the DA outer loop and in forecasts,





and the 60 km mesh is used for analysis increments in the DA inner loop. In the experiments presented, "dual-mesh" is wholly
equivalent to a traditional dual-resolution incremental variational minimization. However, it is also possible to use variable
resolution meshes in either the inner or outer loop in JEDI-MPAS such that there may be many more than 2 mesh spacings
(resolutions). *EDA* was limited to 1 outer iteration and 60 inner iterations for cost savings, but many more iterations are cheap
in deterministic cycling. We apply 2 outer iterations, each with 60 inner iterations, to improve convergence toward observations
with nonlinear operators. The same 1200 km horizontal and 6 km vertical localization lengths scales are applied as in the two
ensemble experiments.

The only difference between the four deterministic experiments is the choice of BEC, which is summarized in Table 2. There
are three pure 3DEnVar experiments, *gefs100*, *dart100*, and *eda100*, that use 100% ensemble BECs based on 20-member GEFS,
80-member *DART*, and 80-member *EDA*, respectively. *gefs100* is equivalent to the *clrama* experiment from Liu et al. (2022).

There is one hybrid 3DEnVar experiment, *eda75c25*, that uses a mixture of 75% *EDA* ensemble and 25% climatological
BEC. The climatological BEC, $\mathbf{B}_c$ is identical to the one described by Jung and et al. (2023). $\mathbf{B}_c$ is trained using the National
Meteorological Center (NMC) method (Parrish and Derber, 1992) with 366 samples of 48 h minus 24 h GFS forecast differ-
ences. The standard deviation of the trained $\mathbf{B}_c$ is scaled by $1/3$, which was determined to be a quasi-optimal tuning justified
by the need to match the 24 h forecast statistics to the 6 h background forecast duration. The horizontal length scales of stream
function and unbalanced velocity potential (see Sect. 2.1 for more description) are scaled by $1/2$ to account for differences
between the sampled correlation vs. separation distance relationship and the Gaspari-Cohn fitting function used in BUMP.
More details can be found in Jung and et al. (2023).

All four experiments are initialized with a 6 h forecast initialized from a GFS analysis valid at 18:00 UTC 14 April 2018,
which was transformed to the MPAS-A 30 km mesh. The results that follow in Sect. 5.2 reflect statistics aggregated across 27
10 d forecasts initialized from 00:00 UTC analyses valid from 18 April 2018 to 14 May 2018. When error bars are shown,
they indicate 95% confidence intervals of those differences, tabulated via bootstrap resampling. Each of the binned RMSE
differences from the 27 forecasts is treated as an independent and identically distributed sample, and they are re-sampled
10,000 times with replacement. The 95% confidence intervals are then obtained by selecting the sample values at the 2.5[th] and
97.5[th] quantiles, referred to as the "percentile method" by Gilleland et al. (2018).

## 5.2 Results

First we present results for the three pure 3DEnVar experiments to evaluate the efficacy of *EDA* ensemble BECs. Our goal is
to interrogate the *EDA* ensemble to determine its utility for future investigations, and not to claim anything about its quality
relative to the other data sources. A low water mark is for forecasts initialized from *eda100* analyses to yield equivalent or
better skill than those initialized from *gefs100* analyses. *dart100* is included to demonstrate how small differences between
*EDA* and *DART* affect deterministic cycling performance. As we described in Sect. 4.2, those differences are not necessarily
related to the ensemble DA algorithms.

Although GEFS was only available as a 20-member product during the period of experimentation presented herein, GEFS
now provides 30-member ensemble forecasts (Zhou et al., 2022). In either case, the GEFS initial conditions are actually 6 h





forecasts initialized from analyses drawn from the 80-member GDAS (Zhou et al., 2017). Therefore, they benefit from being produced by an 80-member EnKF. Additionally, GEFS initial conditions are expected to have added value over JEDI-MPAS's

own ensemble, because they benefit from GDAS's EnKF assimilating many more observation types, having operational-quality observation error and inflation tuning, applying stochastic schemes to ensemble forecasts to treat model error, and being operated at a higher resolution.

Figure 4 shows the *dart100* and *eda100* percent difference of RMSE with respect to GFS analyses compared to the RMSE of the control experiment, *gefs100*, in 0 to 10 day forecasts. *eda100* performs better than *gefs100* with 95% confidence for

$T$, $U$, and $V$. The relative performance of *dart100* with respect to *gefs100* follows similar trends for $T$, $U$, and $V$, slightly outperforming *eda100* for $T$ and slightly underperforming *eda100* for $U$ and $V$. *eda100* produces a nearly neutral impact for $Q_v$, although there is some degradation near the model top that dominates these vertically aggregated errors around day 5.

Figure 5 delves deeper into the largest source of wind observations to impact the DA analyses and subsequent forecasts, satellite AMVs. Although the verification against GFS analyses indicated improvement for both experiments, here the impact

of the *EDA* ensemble in *eda100* is nearly neutral for 1 to 3 day forecasts, and the *DART* ensemble has negative impacts, up to 5% for $U$ at 100 hPa, that are largely limited to tropical latitudes. As described in Sect. 4.2, the observation errors for satellite AMVs have much room for improvement, and the *DART* ensemble has decreasing transient $U$ and $V$ ensemble spread in the tropical free troposphere. More work remains to tune both the observation errors (e.g., following Desroziers et al. (2005)) and BEC (either through spread tuning or covariance and localization improvements) in both the *dart100* and *eda100* experiments,

which may explain some of the poor wind forecast quality.

One observation that is generally consistent with the GFS analyses are sonde $T$ and $U$ in the northern hemisphere (Fig. 6). Both *dart100* and *eda100* achieve up to a 5% positive impact above 50 hPa at day 1 for $T$ and $U$, which slightly decreases at day 2. That positive $T$ impact is collocated with a large cold bias that the non-GEFS BECs improve upon but do not remove entirely. Neither experiment gives statistically significant improvement in the troposphere compared to *gefs100*. Since the

positive impact to sonde $U$ is largely above 100 hPa, it is not surprising that is not reflected in the AMV verification, since satellite AMVs are located at 100 hPa and below.

That vertical distribution of $T$ and $U$ impacts is corroborated in model-space via 2 d forecast verification versus GFS analyses (Fig. 7). *dart100* and *eda100* yield their largest improvements to $T$ and $U$ (more than 30% locally) in the southern polar stratosphere (approximately above levels 35 to 40). Modest positive impacts are seen at most latitudes in the stratosphere,

consistently across both experiments. However, the tropical degradation in tropospheric $T$ and $U$ for *dart100* is more obvious in this model-space verification. Also, *eda100* has degradation in the northern polar troposphere. The vertical distribution of $T$ impacts in *dart100* and *eda100* largely align with the impacts to GNSSRO refractivity.

GNSSRO refractivity, which is sensitive to $T$ and $Q_v$, observations indicate a positive tropospheric impact from both sets of 80-member BECs, although mostly limited to 40°S and poleward. Figure 8 shows the zonally aggregated *dart100* and *eda100*

percent difference of RMSE with respect to refractivity compared to RMSE of *gefs100* for 90°S to 30°S. Both sensitivity experiments give a two-peak statistically significant positive impact centered near 15 km and 22 km altitudes, with a dip near the tropopause that might be explained by its mis-representation in DA background states. *eda100* and *dart100* achieve at




least a 15% positive impact at some stratosphere altitudes out to day 3. In the troposphere, both experiments give comparable positive impacts, over 10% at day 1 and declining slightly with lead time. All positive impacts in the southern hemisphere are
above the planetary boundary layer. Although not shown in Fig. 8, the stratospheric impact of the 80-member BECs persists out to day 5.

It is useful to further explore the impacts of using a hybrid BEC, because that algorithmic enhancement is readily adaptable to future EDA applications. Therefore, we first compare *eda75c25* to *eda100* to demonstrate the benefits of additionally accounting for climatological BECs, and then to *gefs100* to show the total impact of developments herein and in Jung and et al.
415 (2023).

Adding the 25% climatological BEC component and scaling the 80-member ensemble BEC component by 75% adds significant value across a wide range of observation- and model-space verification metrics. Figure 9 shows the $T$, $Q_v$, $U$, and $V$ RMSE with respect to GFS analyses at 2 d lead time for *eda100* and the corresponding percent differences in RMSE for *eda75c25*. Most of the positive impacts for $T$ are limited to the troposphere (below model levels 35-40), and reach up to 14%
above the North Pole. Since the *EDA* ensemble caused some degradation near the North Pole, and did not cause much improvement outside the southern extratropical to polar free troposphere and stratosphere, this is a welcome benefit. The *eda75c25* 2 d forecast moisture also aligns better with GFS analyses than *eda100*, although this also makes up for some degradation in stratospheric $Q_v$ between *eda100* and *gefs100*. Overall, *eda75c25* consistently improves the 2 d wind forecasts throughout the troposphere. It is clear that the climatological BEC makes up for information that is lacking in the completely flow-dependent
ensemble BEC.

Finally, we consider a metric which incorporates information from multiple variables. According to Krishnamurti et al. (2003), a 500 hPa geopotential height anomaly correlation coefficient (ACC) "...greater than 0.6 generally implies that troughs and ridges at 500 hPa are beginning to be properly placed in that forecast." We calculate anomalies with respect to the climatology derived from the 1980-2010 NCEP/NCAR reanalysis products (Kalnay et al., 1996). We conduct cold-start forecasts
initialized from GFS analyses as a reference. The cold-start forecast anomalies are compared to those of GFS analyses. For the *eda75c25* and *gefs100* experiments, their forecast anomalies are compared to those of their respective analyses. Figure 10 shows the 1-to-7 day forecast ACC scores for *eda75c25*, *gefs100*, and the cold-start forecasts. The *gefs100* forecasts are approximately 0.6 days behind the GFS analyses at $0.7 < \text{ACC} < 0.9$. *eda75c25* provides approximately 0.1 to 0.2 days enhanced predictability for for 4 to 6 day forecasts globally.

The *EDA* ensemble-based BEC and the climatological BEC contribute complementary information in the deterministic cycling. Although there are still many avenues for future improvements, these BECs are ready for the non-developer user base. On the basis of this work, JEDI-MPAS has the means to produce an ensemble of forecasts without relying on an external analysis system at each cycle.



## 6    Conclusions and future outlook

This paper has documented the implementation of En-3DEnVar for JEDI-MPAS, and demonstrated its use in both ensemble cycling and in experiments in which a previously computed EDA ensemble provides the BEC for cycling EnVar.

Our cycling DA experiments previously required ensembles of initial conditions from an external system (GEFS) (Liu et al., 2022). Using EDA gives ensemble initial conditions that are consistent with the configuration of MPAS (for example, including a set of hydrometeors consistent with the chosen microphysical scheme) and with the observing network used. EDA 445   also performs better in most scores relative to our previous approach, though with increased computational cost. As a further check on the implementation of EDA, we compared against an experiment using MPAS-DART?a more mature ensemble data-assimilation system?for ensemble BEC generation and found comparable performance in terms of ensemble spread, ensemble mean background RMSE, and subsequent deterministic cycling forecast quality.

A further refinement, which improves relative to using the EDA in EnVar alone, and across almost all latitudes and heights, 450   is the use of a hybrid BEC that is a weighted sum of the ensemble covariances and the static covariances of Jung and et al. (2023). With this refinement, the system is ready to produce research-quality ensembles for future sensitivity studies that aim to enhance JEDI-MPAS.

A non-standard element of our experiments is the use of self-exclusion, as in Bowler et al. (2017a) and **?**. In the update for a given member, the ensemble covariance in the BEC is based on the remaining members, with the own member "excluded." 455   Self-exclusion improves the EDA results because it significantly reduces EDA?s bias toward underestimating the analysis spread. We defer further analysis of self-exclusion to a separate paper.

The largest improvements from EDA relative to using GEFS-based BECs are found in temperature and wind in the stratosphere and throughout the southern hemisphere. The improvements in the stratosphere come despite the EDA ensemble being underdispersive there and despite substantial stratospheric temperature biases in both the analyses and forecasts. Sensitivity 460   tests conducted after this study was complete indicate that GNSSRO assimilation needs to be interrogated and improved, especially in the stratosphere. Better use of those observations should reduce stratospheric temperature bias and potentially also yield better correspondence between the ensemble spread and rms errors.

There are several paths to further improvements in the short-range ensemble forecasts produced by the JEDI-MPAS En-3DEnVar. In all ensemble and deterministic experiments, most of the settings for observation error ($\mathbf{R}$) and QC are taken 465   directly from operational-center-specific implementations either in UFO or in GSI and they reflect the characteristic behavior of a different cycling system. Now that full deterministic and ensemble cycling functionality has been demonstrated with JEDI-MPAS, those settings can be robustly analyzed and tuned (e.g., Desroziers et al., 2005). Accounting for model error in the ensemble-forecast step is also a top priority. Finally, additional improvements could be achieved in *EDA* by using at least 2 outer iterations with more total inner iterations or by enabling an En-4DEnVar, both with corresponding increases in cost.

Previous studies (e.g., Lorenc et al., 2017; Buehner et al., 2017) have pointed out the significantly greater computational expense of EDA based on EnVar compared to an EnKF. The same is true here: the EDA algorithm was roughly 4 times more expensive than the DART EAKF algorithm, both with 80 members. The primary drivers of EDA cost are the (1) reading




and storing of 80 ensemble background states by each of those independent *Variational* members, and then (2) performing localization and multiplication for the 80 perturbations in each inner loop iteration; together those account for $\frac{2}{3}$ of the total
cost of the DA step. While recent model-space localization techniques hold promise for maintaining analysis covariance quality with fewer EDA members (i.e., Lorenc, 2017), we did not exercise those here. However, the total number of members strongly impacts both EnKF and EDA cost.

A solution to problem (1) using parallelization strategies across minimization members was proposed by Arbogast et al. (2017). A number of techniques also exist to address problem (2). The multi-mesh minimizations used in deterministic ex-
periments in Sect. 5 can be extended to the EDA. "Mean-Pert" methods (Lorenc et al., 2017; Buehner et al., 2017) reduce computational costs by simplifying the update of ensemble perturbations as compared to that of the ensemble mean. Bowler et al. (2017b) used a Mean–Pert algorithm to realize a factor of 3 cost reduction in their En-4DEnVar (Lorenc et al., 2017). Block EDA algorithms (Mercier et al., 2018) also hold promise for reducing inner iteration count, which Gas (2021) demonstrated in JEDI. Finally, because JEDI is relatively immature, there also remain many opportunities for basic computational
optimization of JEDI-MPAS. For example, after completing the *EDA* and *DART* experiments, a single-precision in-core memory and computation capability was added for JEDI-MPAS states (e.g., $\boldsymbol{x}_b$, $\boldsymbol{x}_a$) and increments (e.g. $\boldsymbol{\delta x}$). This development reduces the cost for the DA step of the *EDA* experiment by 25%

The progress demonstrated herein is a testament to the fact that innovations introduced into JEDI by one contributor (e.g., JCSDA, NOAA, NASA, U.S. Navy, U.S. Air Force, UK Met Office, NCAR) are more easily leveraged by partners than
was previously possible with separate DA software frameworks. Together with Liu et al. (2022) and Jung and et al. (2023), the demonstration of EDA for JEDI-MPAS provides a foundation for more complex endeavors. In particular, variable mesh resolution is one of the main motivations for MPAS-A and has been demonstrated to produce more realistic forecasts than a nested domain near regions of mesh refinement (Park et al., 2014). Work is already under way to demonstrate variable resolution and regional mesh capabilities in JEDI-MPAS.

*Code availability.* JEDI-MPAS 2.0.0-beta is publicly released on GitHub, accessible in the release/2.0.0-beta branch of mpas-bundle (https://github.com/JCSDA/mpas-bundle/tree/release/2.0.0-beta). It is also available from Zenodo at https://doi.org/10.5281/zenodo.7630054 (Joint Center For Satellite Data Assimilation and National Center For Atmospheric Research, 2022). Global Forecast System analysis data are downloaded from NCAR Research Data Archive https://rda.ucar.edu/datasets/ds084.1/ (last access: 21 October 2022; National Centers For Environmental Prediction/National Weather Service/NOAA/U.S. Department Of Commerce, 2015). Global Ensemble Forecast System en-
semble analysis data are downloaded from https://www.ncei.noaa.gov/products/weather-climate-models/global-ensemble-forecast (last access: 21 October 2022). Conventional and satellite observations assimilated are downloaded from https://rda.ucar.edu/datasets/ds337.0/ (last access: 21 October 2022; National Centers For Environmental Prediction/National Weather Service/NOAA/U.S. Department Of Commerce, 2008) and https://rda.ucar.edu/datasets/ds735.0/ (last access: 21 October 2022; National Centers For Environmental Prediction/National Weather Service/NOAA/U.S. Department Of Commerce, 2009).



*Author contributions.* The first author designed, conducted, and analyzed all En-3DEnVar and 3DEnVar experiments, and wrote the manuscript. Zhiquan Liu and Chris Snyder aided with experiment design and analysis. Craig Schwartz designed and conducted the DART cycling experiment, and wrote the description for MPAS-DART and the associated experiment. All co-authors contributed to the development of the JEDI-MPAS source code and experimental workflow, preparation of externally sourced data, design of experiments, and preparation of the manuscript.

*Competing interests.* The authors have no competing interests.

*Acknowledgements.* The National Center for Atmospheric Research is sponsored by the National Science Foundation of the United States. This research has been supported by the United States Air Force (grant no. NA21OAR4310383). We would like to acknowledge high-performance computing support from Cheyenne (doi:10.5065/D6RX99HX) provided by NCAR's Computational and Information Systems Laboratory, sponsored by the National Science Foundation. Michael Duda in the Mesoscale and Microscale Meteorology (MMM) laboratory
provided guidance on modifying MPAS-A for JEDI-MPAS applications.



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





**Figure 1.** Root-mean-square (rms) of ensemble mean background difference to GFS analyses ($\delta x_{GFSa}$) and background ensemble standard deviation ($\sigma_{x_b}$) for model simulated (a) temperature ($T$), (b) water vapor mixing ratio ($Q_v$), and (c) zonal ($U$) and (d) meridional ($V$) wind components, all versus model level. Statistics are tabulated across all grid columns for *DART* and *EDA* experiments from every 6 h ensemble backgrounds between 00:00 UTC 17 April 2018 and 00:00 UTC 14 May 2018.



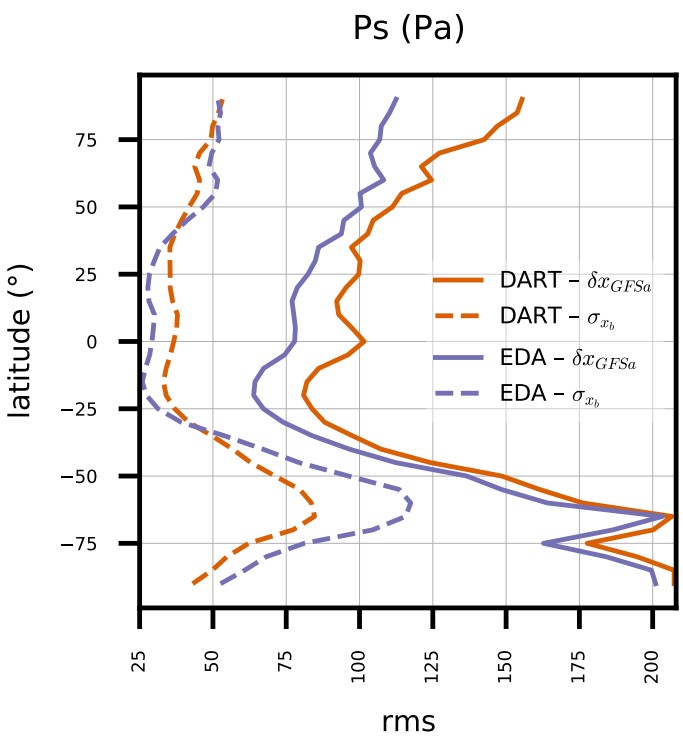

**Figure 2.** Same as Fig. 1, except for model simulated surface pressure ($P_s$) versus latitude.





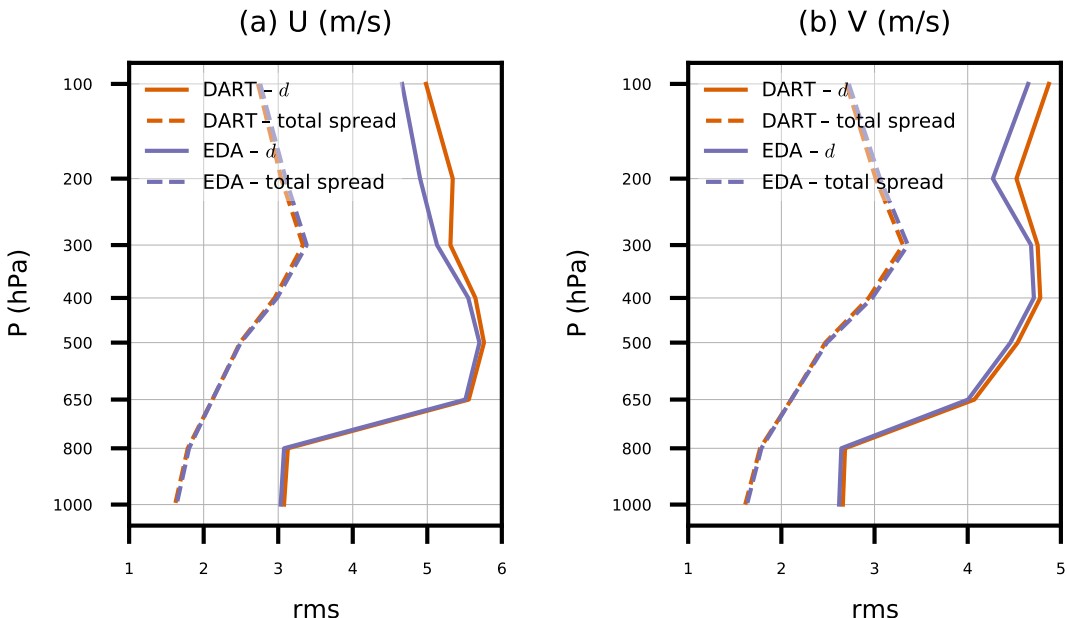

**Figure 3.** Root-mean-square (rms) of ensemble mean background innovation ($d$) and background ensemble total spread for satellite AMV (a) zonal and (b) meridional wind components, versus pressure. Statistics are tabulated globally for *DART* and *EDA* experiments from every 6 h ensemble backgrounds between 00:00 UTC 17 April 2018 and 00:00 UTC 14 May 2018.





**Figure 4.** Percent difference in $\mathrm{rms}(\delta x_{GFSa})$ of *dart100* and *eda100* with respect to $\mathrm{rms}(\delta x_{GFSa})$ of *gefs100* (i.e., $100 \cdot [rms(\delta x_{GFSa})|_{eda100} - rms(\delta x_{GFSa})]|_{gefs100}/rms(\delta x_{GFSa})|_{gefs100}$). Values greater than zero indicate degradation relative to *gefs100*, while values less than zero indicate improvement. Error statistics are aggregated for all model grid columns and levels as a function of forecast lead time (0 to 10 d) for simulated (a) temperature, (b) specific humidity, and (c) zonal and (d) meridional wind components, and pertain to 27 forecasts initialized from 00:00 UTC analyses from 18 April to 14 May 2018. Error bars indicate 95% confidence intervals determined via bootstrap resampling (see text for description).

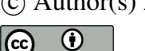

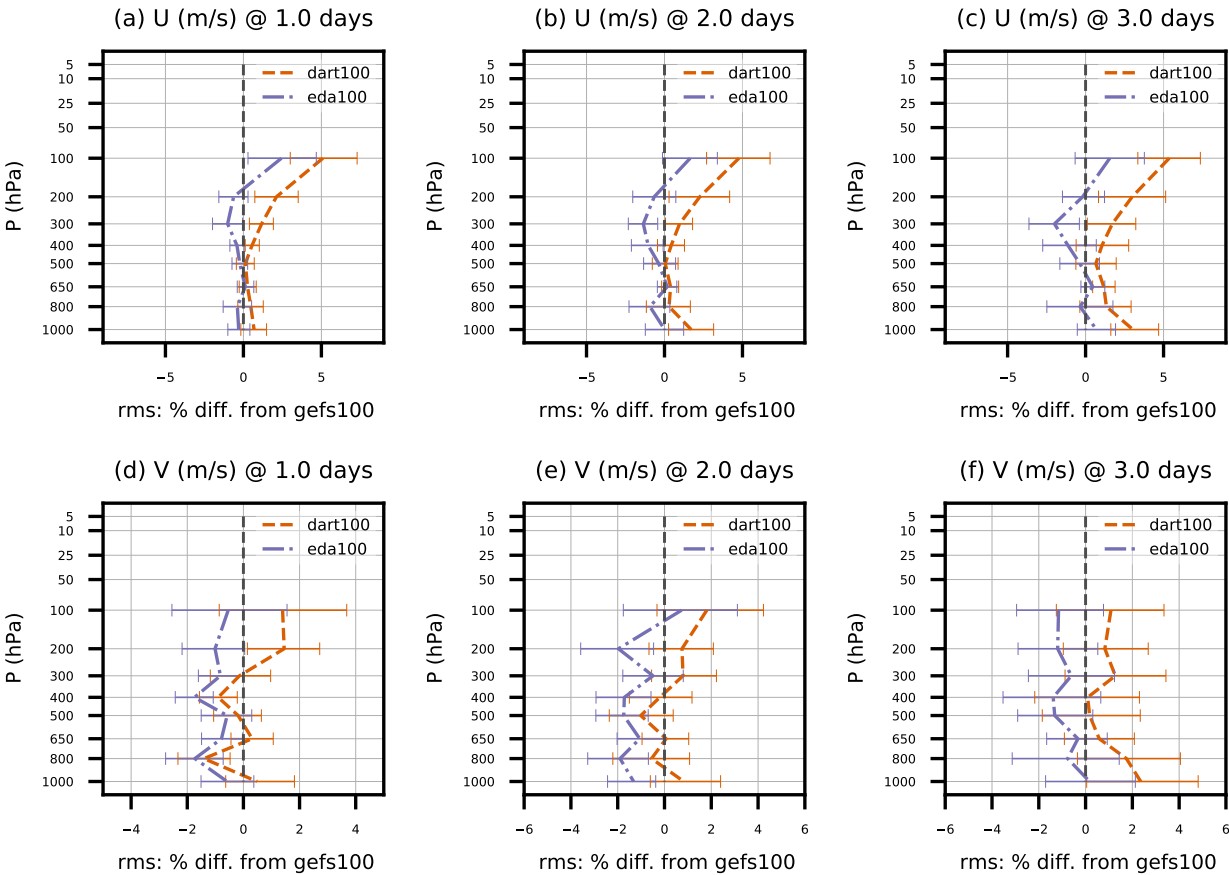

**Figure 5.** Percent difference in rms(O-F) of *dart100* and *eda100* with respect to rms(O-F) of *gefs100* for satellite AMVs. Error statistics are aggregated for individual pressure bins and as a function of forecast lead time (1 to 3 d) for (a-c) zonal and (d-f) meridional wind components, and pertain to 27 forecasts initialized from 00:00 UTC from 18 April to 14 May 2018. Error bars indicate 95% confidence intervals determined via bootstrap resampling (see text for description).





**Figure 6.** Same as Fig. 5, except for 1 to 2 d forecasts of sonde (a-b) temperature and (c-d) zonal wind component between 30°N and 90°N.



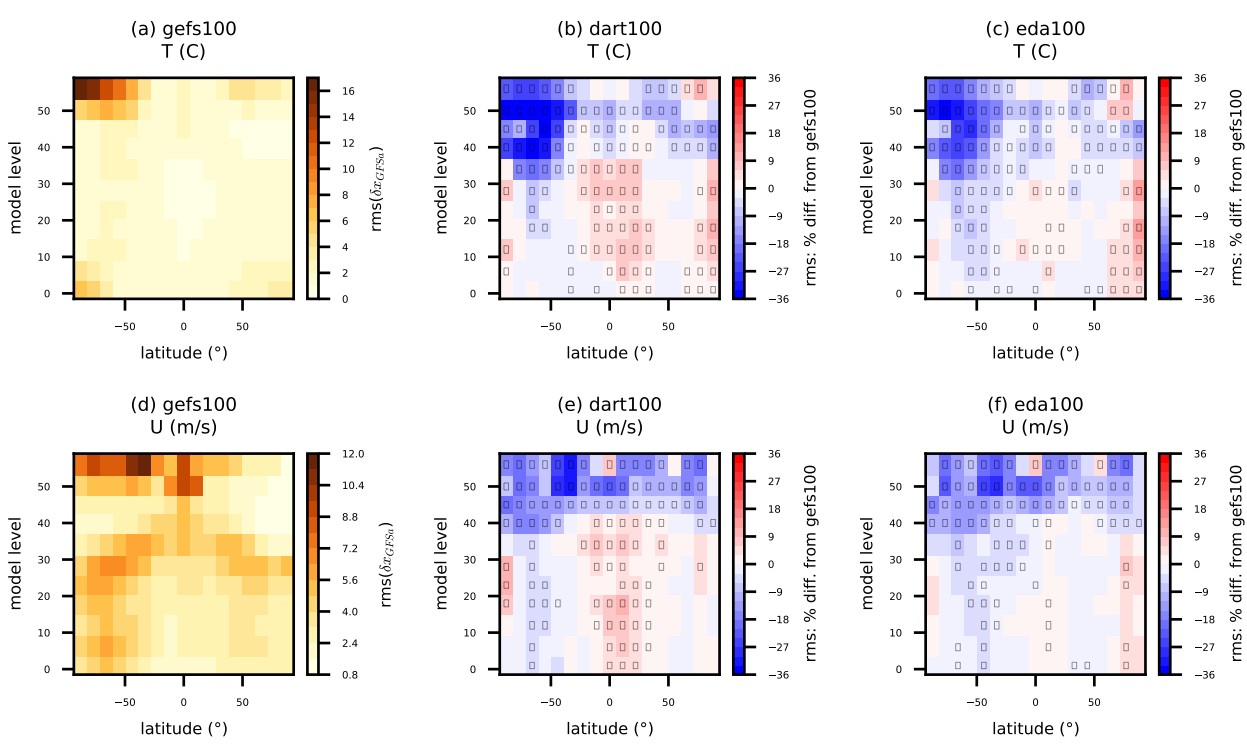

**Figure 7.** (a,d) Reference rms($\delta x_{GFSa}$) for *gefs100* and percent difference of rms($\delta x_{GFSa}$) for (b,e) *dart100* and (c,f) *eda100* at 2 d lead time. Statistics are binned in groups of ~5 model levels and 11° latitude bands for simulated (a,b,c) temperature and (d,e,f) zonal wind component, and pertain to 27 forecasts initialized from 00:00 UTC from 18 April to 14 May 2018. Inset black rectangles in individual bins indicate that the difference of rms between experiments is nonzero with at least 95% confidence, as determined via bootstrap resampling (see text for description).



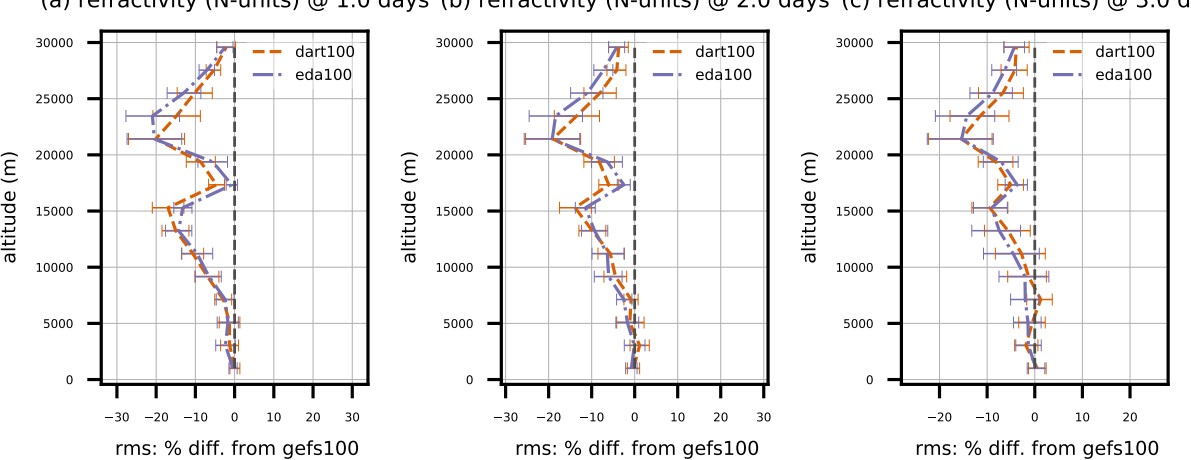

**Figure 8.** Same as Fig. 5, except for GNSSRO refractivity versus altitude between 90°S and 30°S. Error bars indicate 95% confidence intervals determined via bootstrap resampling (see text for description).



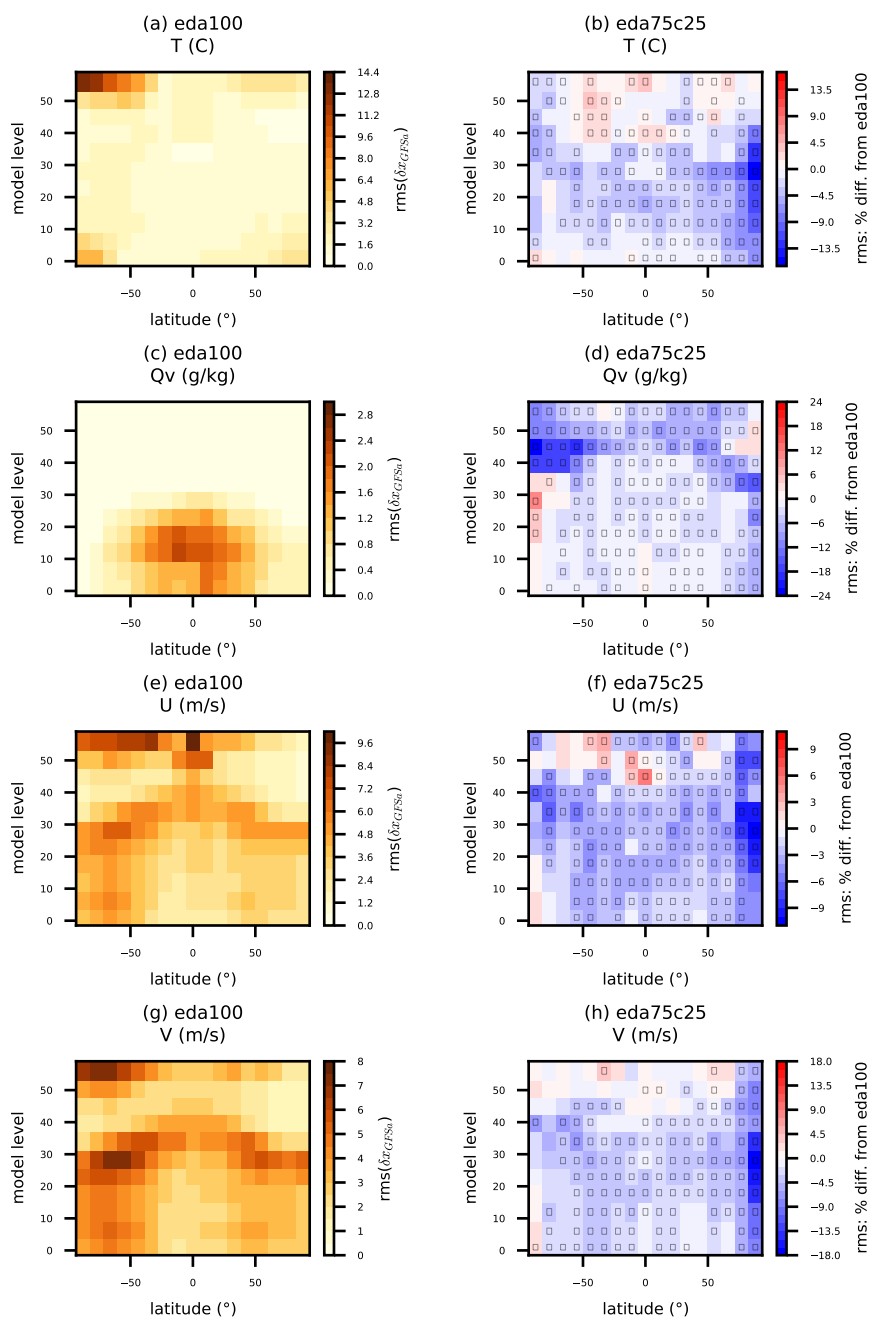

**Figure 9.** Same as Fig. 7, except that the (a,c,e,g) reference rms is for *eda100* and (b,d,f,h) percent difference is shown for *eda75c25*. Statistics are tabulated for simulated (a,b) temperature, (c,d) water vapor mixing ratio, and (e,f) zonal and (g,h) meridional wind components.

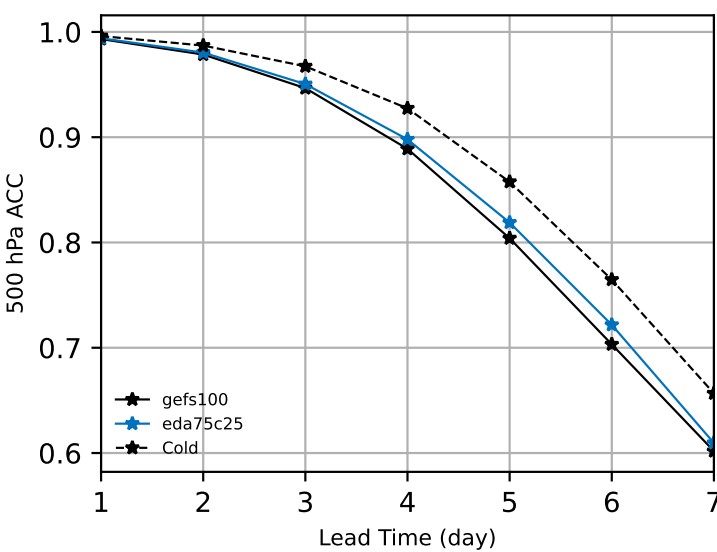

**Figure 10.** 1 to 7 d Anomaly Correlation Coefficient (ACC) of 500 hPa geopotential height for *gefs100*, *eda75c25*, and cold-start forecasts initialized from GFS analyses (Cold). Statistics are tabulated over all grid cells and pertain to 17 forecasts initialized from 00:00 UTC from 18 April to 4 May 2018.





**Table 1.** Observation error distribution versus pressure for satellite AMVs.

| P (hPa) | 1000 | 950 | 800 | 650 | 600 | 550 | 500 | 450 | 400 | 350 | 300 | 250 | 200 | 150 | 100 |
|---|---|---|---|---|---|---|---|---|---|---|---|---|---|---|---|
| $\sigma_o$ (m s$^{-1}$) | 1.4 | 1.5 | 1.6 | 1.8 | 1.9 | 2.0 | 2.1 | 2.3 | 2.6 | 2.8 | 3.0 | 3.2 | 2.7 | 2.4 | 2.1 |





**Table 2.** Deterministic 3DEnVar experiments

| Experiment | BEC |
| --- | --- |
| *gefs100* | 100% 20-member ensemble; 6 h 60 km MPAS-A forecasts from GEFS analyses |
| *dart100* | 100% 80-member ensemble; DART background forecasts |
| *eda100* | 100% 80-member ensemble; EDA background forecasts |
| *eda75c25* | 75% EDA background forecasts, 25% climatological |