# Peer review of "Data assimilation for the Model for Prediction Across Scales – Atmosphere with the Joint Effort for Data assimilation Integration (JEDI-MPAS 2.0.0-beta): ensemble of 3D ensemble-variational (En-3DEnVar) assimilations"

_Geoscientific Model Development, 2023_

## Author Response (AR1)

**Itemized responses to RC1 from anonymous reviewer:**

**RC** - L80: Setting the sample mean...

**AC -** added sentence: "Removing the perturbation bias is standard practice to avoid biasing the posterior, although doing so alters the distribution of the perturbations."

**RC** - L152: Here and below the word "variational" appears in italics…

**AC** - When "*Variational"* (Capitilized and *italics*) we are referring to the JEDI-MPAS application that conducts variational data assimilation (as indicated in L152).  Identical syntax is used for *HofX3D* on L274.

**RC** - L249: Running an NWP experiment for a short period (1 month) can lead to issues with the results…

**AC** - added sentence: "This 1-month experimental period is too short to draw broad conclusions, but is sufficient to demonstrate the new En-3DEnVar capability."

**RC** - L265: Not only is a higher-resolution model more accurate than a lower-resolution model, it is also typically more active…

**AC** - added clause: "and resolved physics are more active"

**RC** - L272: A comparison of model- and observation-space localization was conducted by Greybush et al, MWR (2011) (https://doi.org/10.1175/2010MWR3328.1).  This should be referenced.  It should also be noted that they concluded that length-scales for observation-space localization should be shorter than those for model-space localization…

**AC** - Thank you for this insight and citation.  Sentence and citation added on L272: "Although Greybush et al (2011) found that optimal observation-space localization lengths ought to be shorter than those in model-space, careful tuning of the *DART* experiment was not our focus."  We also modified paragraph starting on L279 for additional transparency, which now ends as "That difference, coupled with differences in algorithm, localization, and inflationary measures between *EDA* and *DART* (Sect. 2), mean that throughout the month of cycling the two experiments assimilate slightly different observations, because different observations could fail the background check between experiments. Nonetheless, any differences in assimilated observations reflect differences in the assimilation algorithms and their particular settings rather than the observation rejection."

**RC** - L273: The value of RTPS being used (1) is shocking…

**AC** - As stated previously, the tuning of the DART experiment was not our focus. The original RTPS paper (Whitaker and Hamill, 2012, https://doi.org/10.1175/MWR-D-11-00276.1) found optimal inflation factors less than 1 and Bowler et al. (2007, https://doi.org/10.1002/qj.3004) found optimal values larger than 1.  We do not claim to have chosen an optimal value, nor do we claim that the DART experiment is the best it could be in comparison to the EDA experiment.  Setting the RTPS inflation factor to 1 simply means that the background spread is declared sufficient.  It is likely that without stochastic perturbations during the integration and additional inflation that is not the case. We already mention the need for model error accounting in our methods and conclusion.

**RC** - L309: The authors conclude that the regional differences in RMSE are directly related to the differences in spread between the ensembles in those regions.  It is not clear from this short discussion whether that is a valid connection to make.  Thus the authors should either provide evidence of the connection, or remove this sentence.

**AC** - Good point.  Modified causal to correlative relationships with the underlined words in the following: "In separate diagnostics that further narrow down...*DART*'s larger local spread correlates with lower RMSE in the western tropical Pacific Ocean and *EDA*'s smaller local spread correlates with lower RMSE...Pacific Ocean.  Those correlations could be associated ..."

**RC** - L335: Section 5 needs to begin with a clear description of the deterministic experiments which are being proposed, but this appears to be missing.  Perhaps the authors could note that they are running deterministic 3DEnVar experiments following in the footsteps of Liu et al (2022).  It should be stated whether the ensembles in the previous section are run prior to the experiments in section 5, or whether they are "coupled" to the DA experiments in any way.

**AC** - The first paragraph of section 5 is edited to make more clear that the ensemble and deterministic experiments are run independently.  There is no coupling, such as ensemble recentering.  We feel that the second and third paragraphs (unedited from original) clearly state the differences between the four experiments.

**RC** - L362: Error bars are estimated by a bootstrap resampling.  The authors use 10,000 samples of 27 values.  Using a large number of samples cannot mask the limitations of using a short period of the experiment.  Either here or elsewhere the authors should acknowledge the short length of the trial, and note that the confidence intervals will not be fully representative of the uncertainty in the result (for instance no account is made of seasonal variability of the results).

**AC** - Add two sentences at the end of that paragraph: "It should be noted that the number of samples used here is small, and does not account for seasonal variation in forecast quality. A more robust approach is needed when deploying an operational system."

**RC** - L378: Figure 4 shows the RMSE difference averaged over all horizontal locations and vertical levels. Averaging in the vertical is a very strange thing to do, as the variability of certain variables will change substantially with height. Comparing with the results in Figure 7, which show improvements in T and U in the upper model levels I would assume that this weighting is towards those heights. Given the unusual nature of the metric the authors should at least acknowledge this issue. Perhaps better would be to promote Figure 7 to the front of the discussion, since this graphic seems to be the most informative.

**AC** - If the order of magnitude of the RMSE does not change much across levels, then Figure 4 is relevant. It is therefore less relevant for Qv, but we feel that it does not detract much from the discussion. While Figure 7 gives more insight, it is beneficial to look at aggregated statistics first to get a sense of the primary differences in performance between experiments before diving into the detailed information. We feel this is a question of style of presentation of material, and not something that needs to be edited.

**RC** - L403: Refractivity is a function of pressure, temperature and humidity. Why is pressure not mentioned here in relation to GNSS-RO observations?

**AC** - The reviewer is absolutely correct. However, only surface pressure observations are assimilated, because 3D pressure is not an analyzed variable in the JEDI-MPAS *Variational* application at this time. In fact, 3D pressure is a diagnostic and not prognostic variable in MPAS-A, such that the analyzed and forecast 3D P is a function of the other analyzed variables (including surface pressure). Pressure is held constant in the tangent/adjoint refractivity operator when applied in JEDI-MPAS.

**RC** - L444: Please add "The" in front of EDA.

**AC** - Done.

**RC** - L446: Here and below (L447, L453 and L445) question marks appear in the text, which looks like characters which have not been represented correctly. Please correct these errors.

**AC** - Done.

**RC** - L477: Please add "the" in front of EnKF.

**AC** - Done.

**RC** - Figure 2: Please could the x-axis start at zero, as I initially thought that the EDA spread in the tropics was much smaller than the DART spread (i.e. very close to zero)?

**AC** - Good idea. Done.

**RC** - Table 1: Please change "Obsevation error distribution" to "Observation error standard deviation".

**AC** - Done.

**Itemized responses to CC1 from Lili Lei:**

**RC** - l46, model space localization can also be implemented in ensemble Kalman filters (Bishop et al. 2018; Lei et al. 2018; Lei et al. 2021).

**AC** -  That is a very good point.  We were citing the UK Met Office's reasoning at the time of their writiing, but we should present the full current picture.  We added this sentence: "On the topic of model-space localization, its use was realizedwith EnKF since 2017 (Bishop et al. 2017 [10.1175/mwr-d-17-0102.1]; Lei et al. 2018)."  Which article are you referring to for Lei et al. 2021 that specifically shows a novel EnKF setup/result (compared to other two) with model-space localization?  If it is relevant to the discussion, we would like to include it.

**RC** - l102, since the reference Jung et al. (2023) is not available, it would be helpful to more details for the Bc.

**AC** -  Jung et al. (2023) was submitted to GMD 21 June 2023, but has not yet been given a DOI.  We would like to discuss with the editorial staff the best path forward on citing it in this manuscript.

**RC** - l127, why "the ensemble forecast in JEDI-MPAS do not at present account for model error"? Since an imperfect model is used, model error can be naturally embedded in the forecast.

**AC** -  Thank you for the question.  Put more simply than in the manuscript text, the usual methods of accounting for model error (e.g., perturbing physics tendencies) are not yet implemented in MPAS-A.  Therefore there was not a switch to turn on.  Otherwise we would have used it as you suggest.

**RC** - l187, if the pressure constraint has significant impact on the analyses, it is straightforward to have a mass adjustment to mimic the pressure constraint in EAKF.

**AC** - We the authors discussed using the same pressure constraint in the EAKF as in the Variational minimizations. However, that would have been another avenue for sensitivity experiments and potentially debugging of cycling workflows that we did not think was necessary in order to introduce the EDA capability.

**RC** - l272, since model space localization tends to have broader localization length scale than observation space localization, it might be more appropriate to have a larger localization length scale in EAKF than in 3DEnVAR.

**AC** - The anonymous reviewer gave the opposite opinion (obs-space localization length should be shorter), citing Greybush et al. (2011). If you have an opposing citation, do you mind sharing it please? As stated in response to RC1…sentence and citation added on L272: "Although Greybush et al (2011) found that optimal observation-space localization lengths ought to be shorter than those in model-space, careful tuning of the DART experiment was not our focus." We also modified paragraph starting on L279 for additional transparency, which now ends as "That difference, coupled with differences in algorithm, localization, and inflationary measures between EDA and DART (Sect. 2), mean that throughout the month of cycling the two experiments assimilate slightly different observations, because different observations could fail the background check between experiments. Nonetheless, any differences in assimilated observations reflect differences in the assimilation algorithms and their particular settings rather than the observation rejection."

**RC** - l273, why not use the same inflation method, either RTPS or RTPP, for both EAKF and En3DEnVAR?

**AC** - That choice was made as a result of the EAKF experiment being produced for a separate project that did have additional funding available for tuning of the inflation method. In an ideal experimental setup, more human/compute resources would have been available for tuning.

**RC** - l305, please provide some explanations for the larger U and V errors of EAKF compared to En3DEnVAR. Are these larger errors resulted from the hybrid background error covariances? If yes, is it possible to have an En3DEnVAR with pure ensemble B, since the goal of the section is to demonstrate that En3DEnVAR has similar posteriors to EAKF?

**AC** - En-3DEnVar did not use a hybrid covariance. Only the deterministic cycling experiment eda75c25 used a hybrid covariance. The climatological

covariance coefficient for all other variational experiments was set to zero. We do not know the exact cause for the larger U and V errors in the EAKF, but likely it is caused by lack of tuning, model error accounting, and additional inflationary measures, as we remark throughout the manuscript.

**RC** - l381, this result is inconsistent with Fig 1c. Please provide some explanations.

**AC** -  Fig 1c shows the 6-h forecasted model-space ensemble spread and ensemble mean RMSE with respect to GFS analyses.  Fig 4c shows 0-10 d RMSE differences of the corresponding deterministic cycling experiments (completely independent without ensemble recentering).  As stated in the following paragraph, the U and V spread for the DART experiment decrease slightly across the 1 month of cycling in the tropical upper troposphere.  So while the aggregated spreads over the month are similar between EDA and DART, their temporal behaviors are not identical, possibly explaining why using their 6-hr forecasts as $B_{ens}$ in deterministic cycling would lead to different performances.  We did not show/declare this causal relationship in the text, because we do not have enough results/diagnostics to support it, nor is it a worthwhile avenue of analysis for a paper introducing the JEDI-MPAS En-3DEnvar.  However, it is one possible explanation and worth investigating in future scientific pursuits.

**RC** - l380-410, the main difference between eda100 and dart100 is from the verification against AMV, but not from the other verification metrics. Why?

**AC** -  See response to previous comment.  Most of the AMV forecast quality difference is in the tropics.

**RC** - l445-455, there multiple ?s that are not correctly displayed.

**AC** -  Fixed.

---

## Author Response (AR2)

Itemized response to Topical Editor

**TE - 1. The statement about T impact and GNSSRO refractivity is unclear and clarification is required.**

**a. Line 409-410: "The vertical distribution of T impacts … largely align with the impacts to GNSSRO refractivity." Is this referred to the impact in southern hemisphere or all latitudes? It is not clear whether the degradation is related to the GNSSRO refractivity assimilation. If yes, this should be clearly addressed. I assume that the GNSSRO observations are available globally, not limited in the southern hemisphere.**

**b. Line 467-469: "Sensitivity tests conducted after this study was complete indicate that GNSSRO assimilation needs to be interrogated and improved". This sentence is unclear to me. Have the authors conducted additional sensitivity experiments with the GNSSRO refractivity? Please provide clarification.**

**AC - 1a.** L412 (new line numbers) "The vertical" —> "The vertical and latitudinal"

**AC - 1b.** Modified sentences starting on L470 (new line numbers). The new text says, "We have since conducted multiple sensitivity tests where we assimilate GNSSRO bending angle instead of refractivity, and carefully tune the bending angle observation errors. Those experiments reduce the stratospheric temperature bias significantly, and additional corrective measures are still under investigation."

**TE - 2. I suggest that the authors should briefly comment about the neutral impact of the Qv field in dart100 and eda100, compared with gefs100. While there is clear difference between eda100 and gefs100 in other fields (U, V, T), why shouldn't we expect improvement in the Qv field? Is this related to the optimization of microwave radiance assimilation?**

**AC -** Added two sentences starting on L391 (new line numbers): "The neutral Qv impact in this globally- and vertically-aggregated metric is likely due to the limited assimilation of moisture-sensitive observations (only sondes and aircraft). Assimilating radiances from water vapor channels in all-sky scenes (i.e., Liu et al., 2022) would better reveal the Qv impacts of these experimental ensembles."